# COUNTERINTUITIVE RL: THE HIDDEN VALUE OF ACTING BAD

## ABSTRACT

Learning to make sequential decisions solely from interacting with an environment without any supervision has been achieved by the initial installation of deep neural networks as function approximators to represent and learn a value function in high-dimensional MDPs. Reinforcement learning policies face exponentially growing state spaces in experience collection in high dimensional MDPs resulting in a dichotomy between computational complexity and policy success. In our paper we focus on the agent's interaction with the environment in a high-dimensional MDP during the learning phase and we introduce a theoretically-founded novel method based on experiences obtained through extremum actions. Our analysis and method provides a theoretical basis for effective, accelerated and efficient experience collection, and further comes with zero additional computational cost while leading to significant acceleration of training in deep reinforcement learning. We conduct extensive experiments in the Arcade Learning Environment with high-dimensional state representation MDPs. We demonstrate that our technique improves the human normalized median scores of Arcade Learning Environment by $248\%$ in the low-data regime.

## 1 INTRODUCTION

Utilization of deep neural networks as function approximators enabled learning functioning policies in high-dimensional state representation MDPs (Mnih et al., 2015). Following this initial work, the current line of work trains deep reinforcement learning policies to solve highly complex problems from game solving (Hasselt et al., 2016; Schrittwieser et al., 2020) to designing algorithms (Mankowitz et al., 2023). Yet there are still remaining unsolved problems restricting the current capabilities of deep neural policies. One of the main intrinsic open problems in deep reinforcement learning research is sample complexity and experience collection in high-dimensional state representation MDPs. While prior work extensively studied the policy's interaction with the environment in bandits and tabular reinforcement learning, and proposed various algorithms and techniques optimal to the tabular form or the bandit context (Fiechter, 1994; Kearns & Singh, 2002; Brafman & Tennenholtz, 2002; Kakade, 2003; Lu & Roy, 2019), experience collection in deep reinforcement learning remains an open challenging problem while practitioners repeatedly employ quite simple yet effective techniques (i.e. $\epsilon$-greedy) (Whitehead & Ballard, 1991; Flennerhag et al., 2022; Hasselt et al., 2016; Wang et al., 2016; Hamrick et al., 2020; Kapturowski et al., 2023).

Despite the provable optimality of the techniques designed for the tabular or bandit setting, they generally rely strongly on the assumptions of tabular reinforcement learning, and in particular on the ability to record tables of statistical estimates for every state-action pair which have size growing with the number of states times the number of actions. Hence, these assumptions are far from what is being faced in the deep reinforcement learning setting where states and actions can be parametrized by high-dimensional representations. Thus, in high-dimensional complex MDPs, for which deep neural networks are used as function approximators, the efficiency and the optimality of the methods proposed for tabular settings do not transfer well to deep reinforcement learning experience collection (Kakade, 2003). Hence, in deep reinforcement learning research still, naive and standard techniques (e.g. $\epsilon$-greedy) are preferred over both the optimal tabular techniques and over the particular recent experience collection techniques targeting only high scores for particular games (Mnih et al., 2015; Hasselt et al., 2016; Wang et al., 2016; Anschel et al., 2017; Bellemare et al., 2017; Dabney et al., 2018; Lan et al., 2020; Flennerhag et al., 2022; Kapturowski et al., 2023).

Sample efficiency in deep neural policies still remains to be one of the main challenging problems restricting research progress in reinforcement learning. The magnitude of the number of samples required to learn and adapt continuously is one of the main limiting factors preventing current state-of-the-art deep reinforcement learning algorithms from being deployed in many diverse settings, but most importantly one of the main challenges that needs to be dealt with on the way to building neural policies that can generalize and adapt continuously in non-stationary environments. In our paper we aim to seek answers for the following questions:

- *How can we construct policies that can collect unique experiences in a high-dimensional state representation MDP without any additional cost?*
- *What is the natural theoretical motivation that can be used to design a zero-cost experience collection strategy while achieving high sample efficiency?*

To be able to answer these questions, in our paper we focus on environment interactions in deep reinforcement learning and make the following contributions:

- We propose a fundamental theoretically well-motivated improvement to temporal difference learning based on state-action value function minimization that increases the information gain from the environment interactions of the policy in a given MDP.
- We conduct an extensive study in the Arcade Learning Environment 100K benchmark with the state-of-the-art algorithms and demonstrate that our temporal difference learning algorithm improves performance by $248\%$ across the entire benchmark compared to the baseline algorithm.
- We demonstrate the efficacy of our proposed MaxMin TD Learning algorithm in terms of sample-efficiency. Our method based on maximizing novel experiences via minimizing the state-action value function reaches approximately to the same performance level as model-based deep reinforcement learning algorithms, without building and learning any model of the environment.
- Finally, from the fact that MaxMin TD learning is a fundamental improvement over canonical methods, our paper demonstrates that any algorithm that uses temporal difference learning can be immediately and simply switched to MaxMin TD learning.

## 2 BACKGROUND AND PRELIMINARIES

The reinforcement learning problem is formalized as a Markov Decision Process (MDP) (Puterman, 1994) $\mathcal{M} = \langle \mathcal{S}, \mathcal{A}, r, \gamma, \rho_0, \mathcal{T} \rangle$ that contains a continuous set of states $s \in \mathcal{S}$, a set of discrete actions $a \in \mathcal{A}$, a probability transition function $\mathcal{T}(s, a, s')$ on $\mathcal{S} \times \mathcal{A} \times \mathcal{S}$, discount factor $\gamma$, a reward function $r(s, a) : \mathcal{S} \times \mathcal{A} \to \mathbb{R}$ with initial state distribution $\rho_0$. A policy $\pi(s, a) : \mathcal{S} \times \mathcal{A} \to [0, 1]$ in an MDP assigns a probability distribution over actions for each state $s \in \mathcal{S}$. The main goal in reinforcement learning is to learn an optimal policy $\pi$ that maximizes the discounted expected cumulative rewards $\mathcal{R} = \mathbb{E}_{a_t \sim \pi(s_t, \cdot), s' \sim \mathcal{T}(s, a, \cdot)} \sum_t \gamma^t r(s_t, a_t)$. In $Q$-learning (Watkins, 1989; Watkins & Dayan, 1992) the learned policy is parameterized by a state-action value function $Q : \mathcal{S} \times \mathcal{A} \to \mathbb{R}$, which represents the value of taking action $a$ in state $s$. The optimal state-action value function is learnt via iterative Bellman update

$$Q(s_t, a_t) = r(s_t, a_t) + \gamma \sum_{s_t} \mathcal{T}(s_t, a_t, s_{t+1}) \mathcal{V}(s_{t+1}).$$

where $\mathcal{V}(s_{t+1}) = \max_a Q(s_{t+1}, a)$. Let $a^*$ be the action maximizing the state-action value function, $a^*(s) = \arg\max_a Q(s, a)$, in state $s$. Once the $Q$-function is learnt the policy is determined via taking action $a^*(s) = \arg\max_a Q(s, a)$. Temporal difference improves the estimates of the state-action values in each iteration via the Bellman Operator (Bellman, 1957)

$$\Omega^\pi Q(s, a) = \mathbb{E}_{a_t \sim \pi(s_t, \cdot), s' \sim \mathcal{T}(s, a, \cdot)} \sum_t \gamma^t r(s_t, a_t) + \gamma \mathbb{E}_{a \sim \pi(s, \cdot), s' \sim \mathcal{T}(s, a, \cdot)} \max_{a'} Q(s, a')$$

For distributional reinforcement learning, QRDQN is an algorithm that is based on quantile regression (Koenker & Hallock, 2001; Koenker, 2005) temporal difference learning

$$\Omega \mathcal{Z}(s, a) = r(s, a) + \gamma \mathcal{Z}(s', \arg\max_{a'} \mathbb{E}_{z \sim \mathcal{Z}(s', a')}[z]) \text{ and } \mathcal{Z}(s, a) := \frac{1}{N} \sum_{i=1}^{N} \delta_{\theta_i(s, a)}$$

where $\mathcal{Z}_\theta \in \mathcal{Z}_Q$ maps state-action pairs to a probability distribution over values. In deep reinforcement learning, the state space or the action space is large enough that it is not possible to learn and store the state-action values in a tabular form. Thus, the $Q$-function is approximated via deep neural networks.

$$\theta_{t+1} = \theta_t + \alpha(r(s_t, a_t) + \gamma Q(s_{t+1}, \arg\max_a Q(s_{t+1}, a; \theta_t); \theta_t) - Q(s_t, a_t; \theta_t))\nabla_{\theta_t} Q(s_t, a_t; \theta_t)$$

In deep double-$Q$ learning, two $Q$-networks are used to decouple the $Q$-network deciding which action to take and the $Q$-network to evaluate the action taken $\theta_{t+1} = \theta_t + \alpha(r(s_t, a_t) + \gamma Q(s_{t+1}, \arg\max_a Q(s_{t+1}, a; \theta_t); \hat{\theta}_t) - Q(s_t, a_t; \theta_t))\nabla_{\theta_t} Q(s_t, a_t; \theta_t)$. Current deep reinforcement learning algorithms use $\epsilon$-greedy during training (Wang et al., 2016; Mnih et al., 2015; Hasselt et al., 2016; Hamrick et al., 2020; Flennerhag et al., 2022; Kapturowski et al., 2023). In particular, the $\epsilon$-greedy (Whitehead & Ballard, 1991) algorithm takes an action $a_k \sim \mathcal{U}(\mathcal{A})$ with probability $\epsilon$ in a given state $s$, i.e. $\pi(s, a_k) = \frac{\epsilon}{|\mathcal{A}|}$, and takes an action $a^* = \arg\max_a Q(s, a)$ with probability $1 - \epsilon$, i.e.

$$\pi(s, \arg\max_a Q(s, a)) = 1 - \epsilon + \frac{\epsilon}{|\mathcal{A}|}$$

While a family of algorithms have been proposed based on counting state visitations (i.e. the number of times action $a$ has been taken in state $s$ by time step $t$) with provable optimal regret bounds using the principal of optimism in the face of uncertainty in the tabular MDP setting, yet incorporating these count-based methods in high-dimensional state representation MDPs requires substantial complexity including training additional deep neural networks to estimate counts or other uncertainty metrics. As a result, many state-of-the-art deep reinforcement learning algorithms still use simple, randomized experience collection methods based on sampling a uniformly random action with probability $\epsilon$ (Mnih et al., 2015; Hasselt et al., 2016; Wang et al., 2016; Hamrick et al., 2020; Flennerhag et al., 2022; Kapturowski et al., 2023). In our experiments, while providing comparison against canonical methods, we also compare our method against computationally complicated and expensive techniques such as noisy-networks that is based on the injection of random noise with additional layers in the deep neural network (Hessel et al., 2018) in Section 5, and count based methods in Section 4 and Section 6. Note that our method is a fundamental theoretically motivated improvement of temporal difference learning. Thus, any algorithm that is based on temporal difference learning can immediately be switched to MaxMin TD learning.

## 3 BOOSTING TEMPORAL DIFFERENCE

In deep reinforcement learning the state-action value function is initialized with random weights (Mnih et al., 2015; 2016; Hasselt et al., 2016; Wang et al., 2016; Schaul et al., 2016; Oh et al., 2020; Schrittwieser et al., 2020; Hubert et al., 2021). Thus, in the early phase of the training the $Q$-function behaves as a random function rather than providing an accurate representation of the optimal state-action values. In particular, early in training the $Q$-function, on average, assigns approximately similar values to states that are similar, and has little correlation with the immediate rewards. Hence, let us formalize these facts on the state-action value function in the following definitions.

**Definition 3.1** ($\eta$-*uninformed*). Let $\eta > 0$. A $Q$-function parameterized by weights $\theta \sim \Theta$ is $\eta$-uninformed if for any state $s \in \mathcal{S}$ with $a^{\min} = \arg\min_a Q_\theta(s, a)$ we have

$$|\mathbb{E}_{\theta \sim \Theta}[r(s_t, a^{\min})] - \mathbb{E}_{a \sim \mathcal{U}(\mathcal{A})}[r(s_t, a)]| < \eta.$$

**Definition 3.2** ($\delta$-*smooth*). Let $\delta > 0$. A $Q$-function parameterized by weights $\theta \sim \Theta$ is $\delta$-smooth if for any state $s \in \mathcal{S}$ and action $\hat{a} = \hat{a}(s, \theta)$ with $s' \sim \mathcal{T}(s, \hat{a}, \cdot)$ we have

$$|\mathbb{E}_{\theta \sim \Theta}[\max_a Q_\theta(s, a)] - \mathbb{E}_{s' \sim \mathcal{T}(s, \hat{a}, \cdot), \theta \sim \Theta}[\max_a Q_\theta(s', a)]| < \delta$$

where the expectation is over both the random initialization of the $Q$-function weights, and the random transition to state $s' \sim \mathcal{T}(s, \hat{a}, \cdot)$.

**Definition 3.3** (*Disadvantage Gap*). For a state-action value function $Q_\theta$ the disadvantage gap in a state $s \in \mathcal{S}$ is given by $\mathcal{D}(s) = \mathbb{E}_{a \sim \mathcal{U}(\mathcal{A}), \theta \sim \Theta}[Q_\theta(s, a) - Q_\theta(s, a^{\min})]$ where $a^{\min} = \arg\min_a Q_\theta(s, a)$.

The following proposition captures the intuition that choosing the action minimizing the state-action value function will achieve an above-average temporal difference when the $Q$-function on average assigns similar maximum values to consecutive states.

**Proposition 3.4.** *Let $\eta, \delta > 0$ and suppose that $Q_\theta(s, a)$ is $\eta$-uninformed and $\delta$-smooth. Let $s_t \in \mathcal{S}$ be a state, and let $a^{min}$ be the action minimizing the state-action value in a given state $s_t$, $a^{min} = \arg\min_a Q_\theta(s_t, a)$. Let $s_{t+1}^{min} \sim \mathcal{T}(s_t, a^{min}, \cdot)$. Then for an action $a_t \sim \mathcal{U}(\mathcal{A})$ with $s_{t+1} \sim \mathcal{T}(s_t, a_t, \cdot)$ we have*

$$\mathbb{E}_{s_{t+1}^{min} \sim \mathcal{T}(s_t, a^{min}, \cdot), \theta \sim \Theta}[r(s_t, a^{min}) + \gamma \max_a Q_\theta(s_{t+1}^{min}, a) - Q_\theta(s_t, a^{min})]$$

$$> \mathbb{E}_{a_t \sim \mathcal{U}, (\mathcal{A}) s_{t+1} \sim \mathcal{T}(s_t, a_t, \cdot), \theta \sim \Theta}[r(s_t, a_t) + \gamma \max_a Q_\theta(s_{t+1}, a) - Q_\theta(s_t, a_t)] + \mathcal{D}(s_t) - 2\delta - \eta$$

*Proof.* Since $Q_\theta(s, a)$ is $\delta$-smooth we have

$$\mathbb{E}_{s_{t+1}^{\min} \sim \mathcal{T}(s_t, a^{\min}, \cdot), \theta \sim \Theta}[\gamma \max_a Q_\theta(s_{t+1}^{\min}, a) - Q_\theta(s_t, a_{\min})]$$

$$> \gamma \mathbb{E}_{\theta \sim \Theta}[\max_a Q_\theta(s_t, a)] - \delta - \mathbb{E}_{\theta \sim \Theta}[Q_\theta(s_t, a_{\min})]$$

$$> \gamma \mathbb{E}_{s_{t+1} \sim \mathcal{T}(s_t, a_t, \cdot), \theta \sim \Theta}[\max_a Q_\theta(s_{t+1}, a)] - 2\delta - \mathbb{E}_{\theta \sim \Theta}[Q_\theta(s_t, a_{\min})]$$

$$\geq \mathbb{E}_{a_t \sim \mathcal{U}(\mathcal{A}), s_{t+1} \sim \mathcal{T}(s_t, a_t, \cdot), \theta \sim \Theta}[\gamma \max_a Q_\theta(s_{t+1}, a) - Q_\theta(s_t, a_t)] + \mathcal{D}(s_t) - 2\delta$$

where the last line follows from Definition 3.3. Further, because $Q_\theta(s, a)$ is $\eta$-uninformed,

$$\mathbb{E}_{\theta \sim \Theta}[r(s_t, a^{\min})] > \mathbb{E}_{a_t \sim \mathcal{U}(\mathcal{A})}[r(s_t, a_t)] - \eta.$$

Combining with the previous inequality completes the proof. □

In words, the proposition shows that the temporal difference achieved by the minimum-value action is above-average by an amount approximately equal to the disadvantage gap. The above argument can be extended to the case where action selection and evaluation in the temporal difference are computed with two different sets of weights $\theta$ and $\hat{\theta}$ as in double $Q$-learning.

**Definition 3.5** ($\delta$-*smoothness for Double-Q*). *Let $\delta > 0$. A pair of $Q$-functions parameterized by weights $\theta \sim \Theta$ and $\hat{\theta} \sim \Theta$ are $\delta$-smooth if for any state $s \in \mathcal{S}$ and action $\hat{a} = \hat{a}(s, \theta) \in \mathcal{A}$ with $s' \sim \mathcal{T}(s, \hat{a}, \cdot)$ we have*

$$\left| \mathbb{E}_{s' \sim \mathcal{T}(s, \hat{a}, \cdot), \theta \sim \Theta, \hat{\theta} \sim \Theta} \left[ Q_{\hat{\theta}}(s, \arg\max_a Q_\theta(s, a)) \right] \right.$$

$$\left. - \mathbb{E}_{s' \sim \mathcal{T}(s, \hat{a}, \cdot), \theta \sim \Theta, \hat{\theta} \sim \Theta} \left[ Q_{\hat{\theta}}(s', \arg\max_a Q_\theta(s', a)) \right] \right| < \delta$$

where the expectation is over both the random initialization of the $Q$-function weights $\theta$ and $\hat{\theta}$, and the random transition to state $s' \sim \mathcal{T}(s, \hat{a}, \cdot)$.

With this definition we can then prove that choosing the minimum valued action will lead to a temporal difference that is above-average by approximately $\mathcal{D}(s)$.

**Proposition 3.6.** *Let $\eta, \delta > 0$ and suppose that $Q_\theta$ and $Q_{\hat{\theta}}$ are $\eta$-uniformed and $\delta$-smooth. Let $s_t \in \mathcal{S}$ be a state, and let $a^{min} = \arg\min_a Q_\theta(s_t, a)$. Let $s_{t+1}^{min} \sim \mathcal{T}(s_t, a^{min}, \cdot)$. Then for an action $a_t \sim \mathcal{U}(\mathcal{A})$ with $s_{t+1} \sim \mathcal{T}(s_t, a_t, \cdot)$ we have*

$$\mathbb{E}_{s_{t+1} \sim \mathcal{T}(s, a, \cdot), \theta \sim \Theta, \hat{\theta} \sim \Theta}[r(s_t, a^{min}) + \gamma Q_{\hat{\theta}}(s_{t+1}^{min}, \arg\max_a Q_\theta(s_{t+1}^{min}, a)) - Q_\theta(s_t, a^{min})]$$

$$> \mathbb{E}_{a_t \sim \mathcal{U}(\mathcal{A}), s_{t+1} \sim \mathcal{T}(s, a, \cdot), \theta \sim \Theta, \hat{\theta} \sim \Theta}[r(s_t, a_t) + \gamma Q_{\hat{\theta}}(s_{t+1}, \arg\max_a Q_\theta(s_{t+1}, a)) - Q_\theta(s_t, a_t)]$$

$$+ \mathcal{D}(s_t) - 2\delta - \eta$$

*Proof.* Since $Q_\theta$ and $Q_{\hat{\theta}}$ are $\delta$-smooth we have

$$\mathbb{E}_{s_{t+1}^{\min} \sim \mathcal{T}(s_t, a^{\min}, \cdot), \theta \sim \Theta, \hat{\theta} \sim \Theta}[+\gamma Q_{\hat{\theta}}(s_{t+1}^{\min}, \arg\max_a Q_\theta(s_{t+1}^{\min}, a)) - Q_\theta(s_t, a^{\min})]$$

$$> \mathbb{E}_{s_{t+1}^{\min} \sim \mathcal{T}(s_t, a^{\min}, \cdot), \theta \sim \Theta, \hat{\theta} \sim \Theta}[+\gamma Q_{\hat{\theta}}(s_t, \arg\max_a Q_\theta(s_t, a)) - Q_\theta(s_t, a^{\min})] - \delta$$

$$> \mathbb{E}_{s_{t+1} \sim \mathcal{T}(s_t, a_t, \cdot), \theta \sim \Theta, \hat{\theta} \sim \Theta}[+\gamma Q_{\hat{\theta}}(s_{t+1}, \arg\max_a Q_\theta(s_{t+1}, a)) - Q_\theta(s_t, a^{\min})] - 2\delta$$

$$\geq \mathbb{E}_{s_{t+1} \sim \mathcal{T}(s_t, a_t, \cdot), \theta \sim \Theta, \hat{\theta} \sim \Theta}[+\gamma Q_{\hat{\theta}}(s_{t+1}, \arg\max_a Q_\theta(s_{t+1}, a)) - Q_\theta(s_t, a_t)] + \mathcal{D}(s_t) - 2\delta$$

where the last line follows from Definition 3.3. Further, because $Q_\theta$ and $Q_{\hat{\theta}}$ are $\eta$-uniformed, $\mathbb{E}_{\theta \sim \Theta, \hat{\theta} \sim \Theta}[r(s_t, a^{\min})] > \mathbb{E}_{a_t \sim \mathcal{U}(\mathcal{A})}[r(s_t, a_t)] - \eta$. Combining with the previous inequality completes the proof. $\square$

> **Core Counterintuition:** *How could minimizing the state-action value function accelerate learning?*

At first, the results in Proposition 3.4 and 3.6 might appear counterintuitive. Yet, understanding this counterintuitive fact relies on first understanding the intrinsic difference between randomly initialized state-action value function, i.e. $Q_\theta$, and the optimal state-action value function, i.e. $Q^*$. In particular, from the perspective of the function $Q^*$, the action $a^{\min}(s) = \arg\min_a Q_\theta(s, a)$ is a uniform random action. However, from the perspective of the function $Q_\theta$, the action $a^{\min}$ is meaningful, in that it will lead to a higher TD-error update than any other action. In fact, Proposition 3.4 and 3.6 precisely provides the formalization that the temporal difference achieved by taking the minimum action is larger than that of a random action by an amount equal to the disadvantage gap $\mathcal{D}(s)$. In order to reconcile these two statements it is useful at this point to look at the limiting case of the $Q$ function at initialization. In particular, the following proposition shows that, at initialization, the distribution of the minimum value action in a given state is uniform by itself, but is constant once we condition on the weights $\theta$.

**Proposition 3.7.** *Let $\theta$ be the random initial weights for the Q-function. For any state $s \in \mathcal{S}$ let $a^{min}(s) = \arg\min_{a' \in \mathcal{A}} Q_\theta(s, a')$. Then for any $a \in \mathcal{A}$*

$$\mathbb{P}_{\theta \sim \Theta}\left[\arg\min_{a' \in \mathcal{A}} Q_\theta(s, a') = a\right] = \frac{1}{|\mathcal{A}|}$$

*i.e. the distribution $\mathbb{P}_{\theta \sim \Theta}[a^{min}(s)]$ is uniform. Simultaneously, the conditional distribution $\mathbb{P}_{\theta \sim \Theta}[a^{min}(s) \mid \theta]$ is constant.*

*Proof.* See supplementary material for the proof. $\square$

This implies that, in states whose $Q$-values have not changed drastically from initialization, taking the minimum action is almost equivalent to taking a random action. However, while the action chosen early on in training is almost uniformly random when only considering the current state, it is at the same time completely determined by the current value of the weights $\theta$. The temporal difference is also determined by the weights $\theta$. Thus while the marginal distribution on actions taken is uniform, the temporal difference when taking the minimum action is quite different than from the case where an independently random action is chosen. In particular, in expectation over the random initialization $\theta \sim \Theta$, the temporal difference is higher when taking the minimum value action than that of a random action as demonstrated in Section 3.

The main objective of our method is to increase the information gained from each environment interaction via taking the actions that minimize the state-action value function. While minimization of the $Q$-function may initially be regarded as counterintuitive, Section 3 provides the exact theoretical justification on how taking actions that minimize the state-action value function results in higher temporal difference for the corresponding state transitions. Note that our method is a fundamental theoretically well motivated improvement on temporal difference learning. Thus, any algorithm in reinforcement learning that is built upon temporal difference learning can be simply switched to

---

**Algorithm 1:** MaxMin TD Learning

---

**Input:** In MDP $\mathcal{M}$ with $\gamma \in (0, 1]$, $s \in \mathcal{S}$, $a \in \mathcal{A}$ with $Q_\theta(s, a)$ function parametrized by $\theta$, $\mathcal{B}$ experience replay buffer, $\epsilon$ dithering parameter, $\mathcal{N}$ is the training learning steps.

Populating Experience Replay Buffer:
    **for** $s_t$ in $e$ **do**
        Sample $\kappa \sim U(0, 1)$
        **if** $\kappa < \epsilon$ **then**
            $a^{min} = \arg\min_a Q(s_t, a)$
            $s_{t+1}^{min} \sim \mathcal{T}(s_t, a^{min}, \cdot)$
            $\mathcal{B} \leftarrow (r(s_t, a^{min}), s_t, s_{t+1}^{min}, a^{min})$
        **else**
            $a^{\max} = \arg\max_a Q(s_t, a)$
            $s_{t+1} \sim \mathcal{T}(s_t, a^{\max}, \cdot)$
            $\mathcal{B} \leftarrow (r(s_t, a^{\max}), s_t, s_{t+1}, a^{\max})$
        **end if**
    **end for**

Learning:
    **for** $n$ in $\mathcal{N}$ **do**
        Sample from replay buffer
        $\langle s_t, a_t, r(s_t, a_t), s_{t+1} \rangle \sim \mathcal{B}$:
        $\mathcal{TD}$ receives update with probability $\epsilon$:
        $\mathcal{TD} = r(s_t, a^{min}) + \gamma \max_a Q(s_{t+1}^{min}, a) - Q(s_t, a^{min})$
        $\mathcal{TD}$ receives update with probability $1 - \epsilon$:
        $\mathcal{TD} = r(s_t, a^{\max}) + \gamma \max_a Q(s_{t+1}, a) - Q(s_t, a^{\max})$
    **end for**
    $\nabla \mathcal{L}(\mathcal{TD})$

---

**MaxMin TD learning.** Algorithm 1 summarizes our proposed algorithm MaxMin TD Learning based on minimizing the state-action value function as described in detail in Section 3. Note that populating the experience replay buffer and learning are happening simultaneously with different rates. TD receives an update with probability $\epsilon$ solely due to the experience collection.

# 4 MOTIVATING EXAMPLE

To truly understand the intuition behind our counterintuitive foundational method we consider a motivating example the chain MDP. In particular, the chain MDP which consists of a chain of $n$ states $s \in \mathcal{S} = \{1, 2, \cdots n\}$ each with four actions. Each state $i$ has one action that transitions the agent up the chain by one step to state $i + 1$, one action that transitions the agent to state 2, one action that transitions the agent to state 3, and one action which resets the agent to state 1 at the beginning of the chain. All transitions have reward zero, except for the last transition returning the agent to the beginning from the $n$-th state. Thus, when started from the first state in the chain, the agent must learn a policy that takes $n - 1$ consecutive steps up the chain, and then one final step to reset and get the reward. For the chain MDP, we compare standard approaches in temporal difference learning in tabular $Q$-

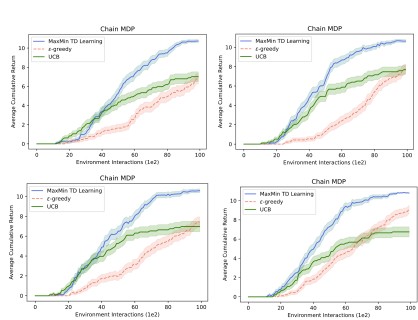

Figure 1: Learning curves in the chain MDP with our proposed algorithm MaxMin TD Learning, the canonical algorithm $\epsilon$-greedy and the UCB algorithm with variations in $\epsilon$.

learning with our method MaxMin TD Learning based on minimization of the state-action values. In particular we compare our method MaxMin TD Learning with both the $\epsilon$-greedy action selection method, and the upper confidence bound (UCB) method. In more detail, in the UCB method the number of training steps $t$, and the number of times $N_t(s, a)$ that each action $a$ has been taken in state $s$ by step $t$ are recorded. Furthermore, the action $a \in \mathcal{A}$ selection is determined as follows:

$$a^{\text{UCB}} = \arg\max_{a \in \mathcal{A}} Q(s, a) + 2\sqrt{\frac{\log t}{N_t(s, a)}}.$$

In a given state $s$ if $N(s, a) = 0$ for any action $a$, then an action is sampled uniformly at random from the set of actions $a'$ with $N(s, a') = 0$. For the experiments reported in our paper the length of the chain is set to $n = 10$. The $Q$-function is initialized by independently sampling each state-action value from a normal distribution with $\mu = 0$ and $\sigma = 0.1$. In each iteration we train the agent using $Q$-learning for 100 steps, and then evaluate the reward obtained by the argmax policy using the current $Q$-function for 100 steps. Note that the maximum achievable reward in 100 steps is 10. Figure

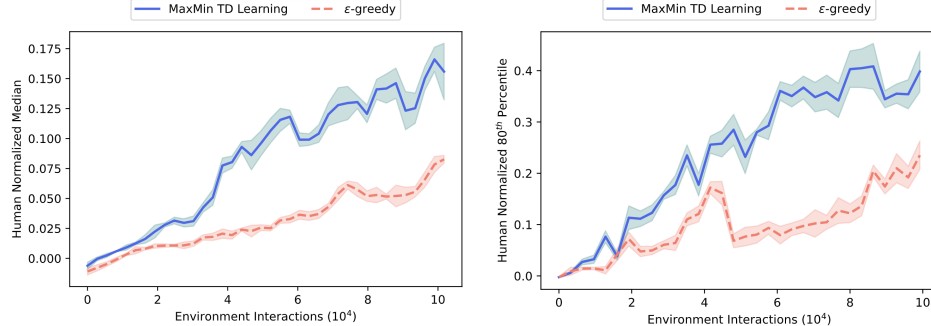

Figure 2: Human normalized scores median and 80[th] percentile over all games in the Arcade Learning Environment (ALE) 100K benchmark for MaxMin TD Learning and the canonical temporal difference learning with $\epsilon$-greedy for QRDQN. Right:Median. Left: 80[th] Percentile.

1 reports the learning curves for each method with varying $\epsilon \in [0.15, 0.25]$ with step size $0.025$. The results in Figure 1 demonstrate that our method converges faster to the optimal policy than either of the standard approaches.

## 5 LARGE SCALE EXPERIMENTAL RESULTS

The experiments are conducted in the Arcade Learning Environment (ALE) (Bellemare et al., 2013). We conduct empirical analysis with multiple baseline algorithms including Double-Q Network (Hasselt et al., 2016) initially proposed by (van Hasselt, 2010) trained with prioritized experience replay (Schaul et al., 2016) without the dueling architecture with its original version (Hasselt et al., 2016), and the QRDQN algorithm that is also described in Section

Table 1: Human normalized scores median, 20[th] and 80[th] percentile across all of the games in the Arcade Learning Environment 100K benchmark for MaxMin TD Learning, $\epsilon$-greedy and NoisyNetworks with DDQN.

| Method | MaxMin TD | $\epsilon$-greedy | NoisyNetworks |
|---|---|---|---|
| Median | **0.0927±0.0050** | 0.0377±0.0031 | 0.0457±0.0035 |
| 20[th] Percentile | **0.0145±0.0003** | 0.0056±0.0017 | 0.0102±0.0018 |
| 80[th] Percentile | **0.3762±0.0137** | 0.2942±0.0233 | 0.1913±0.0144 |

2. The experiments are conducted both in the 100K Arcade Learning Environment benchmark, and the canonical version with 200 million frame training (Mnih et al., 2015; Wang et al., 2016). Note that the 100K Arcade Learning Environment benchmark is an established baseline proposed to measure sample efficiency in deep reinforcement learning research, and contains 26 different Arcade Learning Environment games. The policies are evaluated after 100000 environment interactions. All of the polices in the experiments are trained over 5 random seeds. The hyperparameters and the architecture details are reported in the supplementary material. All of the results in the paper are reported with the standard error of the mean. The human normalized scores are computed as, $\text{HN} = (\text{Score}_{agent} - \text{Score}_{random})/(\text{Score}_{human} - \text{Score}_{random})$. Table 1 reports results of human normalized median scores, 20[th] percentile, and 80[th] percentile for the Arcade Learning Environment 100K benchmark. Furthermore, we also compare our proposed MaxMin TD Learning algorithm with NoisyNetworks as referred to in Section 2. Table 1 further demonstrates that the MaxMin TD Learning algorithm achieves significantly better performance results compared to NoisyNetworks. Primarily, note that NoisyNetworks includes adding layers in the Q-network to increase exploration. However, this increases the number of parameters that have been added in the training process; thus, introducing substantial additional cost. Thus, Table 1 demonstrates that our proposed MaxMin TD Learning algorithm improves on the performance of the canonical algorithm $\epsilon$-greedy by 248% and NoisyNetworks by 204%.

For completeness we also report several results with 200 million frame training (i.e. 50 million environment interactions). In particular, Figure 3 demonstrates the learning curves for our proposed algorithm MaxMin TD Learning and the original version of the DDQN algorithm with $\epsilon$-greedy training (Hasselt et al., 2016). In the large data regime we observe that while in some MDPs our proposed method MaxMin TD Learning that focuses on experience collection with novel temporal difference boosting via minimizing the state-action values converges faster, in other MDPs MaxMin TD Learning simply converges to a better policy. More concretely, while the learning curves of StarGunner, Bowling, JamesBond and BankHeist games in Figure 3 demonstrate the faster conver-

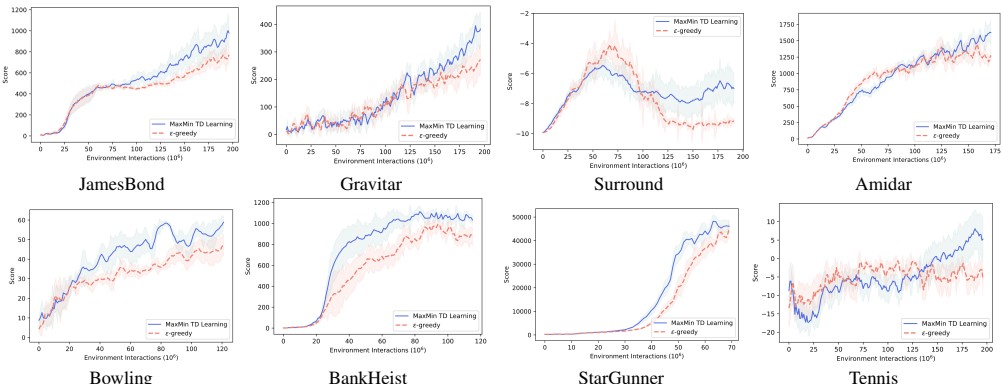

Figure 3: The learning curves of StarGunner, Bowling, Surround, BankHeist, JamesBond, Amidar, Gravitar and Tennis with our proposed method MaxMin TD Learning algorithm and canonical temporal difference learning in the Arcade Learning Environment with 200 million frame training.

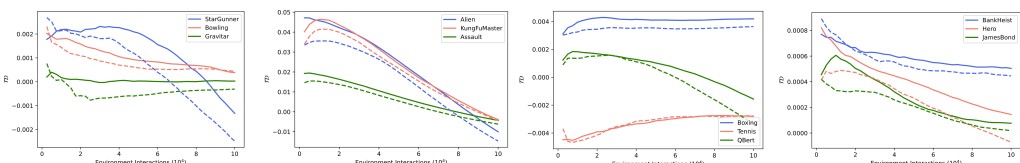

Figure 4: Temporal difference for our proposed algorithm MaxMin TD Learning and the canonical $\epsilon$-greedy algorithm in the Arcade Learning Environment 100K benchmark. Dashed lines report the temporal difference for the $\epsilon$-greedy algorithm and solid lines report the temporal difference for the MaxMin TD Learning algorithm. Colors indicate games.

gence rate of our proposed algorithm MaxMin TD Learning, the learning curves of the JamesBond, Amidar, BankHeist, Surround, Gravitar and Tennis games demonstrate that our experience collection technique not only increases the sample efficiency in deep reinforcement learning, but also results in learning a policy that is more close to optimal compared to learning a policy with the original method used in the DDQN algorithm.

We further compare our proposed MaxMin TD Learning algorithm with another baseline algorithm double-Q learning. In particular, while Figure 5 reports results for double Q-learning, Figure 2 reports results of human normalized median scores and $80^{\text{th}}$ percentile over all of the games of the Arcade Learning Environment (ALE) in the low-data regime for QRDQN. The results reported in Figure 2 once more demonstrate that the performance obtained by the MaxMin TD Learning algorithm is approximately double the performance achieved by the canonical experience collection techniques. The large scale experimental analysis further discovers that the MaxMin TD Learning algorithm achieves substantial sample-efficiency with zero-additional cost across many algorithms and different sample-complexity regimes over canonical baseline alternatives.

## 6    INVESTIGATING THE TEMPORAL DIFFERENCE

The original justification for collecting experiences with the minimum $Q$-value action, is that taking this action tends to result in transitions with higher temporal difference. The theoretical analysis from Proposition 3.4 indicates that, when the $Q$ function is $\delta$-smooth and $\eta$-uninformed, taking the minimum value action results in an increase in the temporal difference proportional to the disadvantage gap. In particular, Proposition 3.4 states that the temporal difference achieved when taking the minimum $Q$-value action in state $s$ exceeds the average temporal difference over a uniform random action by $\mathcal{D}(s) - 2\delta - \eta$. In this section we will investigate the temporal difference and provide empirical measurements of the temporal difference. To measure the change in the temporal difference when taking the minimum action versus the average action, we compare the temporal difference obtained by MaxMin TD Learning with that obtained by $\epsilon$-greedy-based temporal difference learning. In more detail, during training, for each batch $\Lambda$ of transitions of the form $(s_t, a_t, s_{t+1})$ we record,

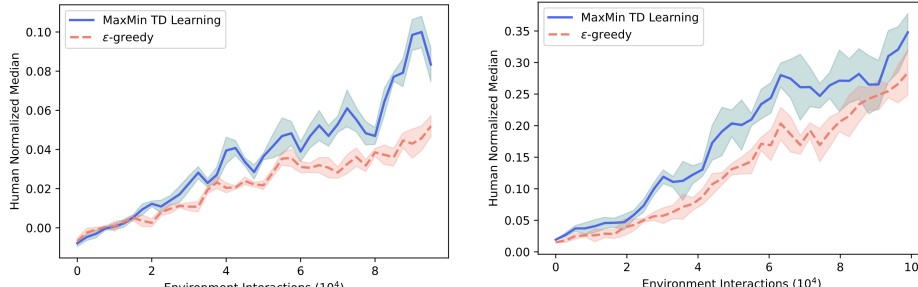

Figure 5: Human normalized scores median and 80th percentile over all games in the Arcade Learning Environment (ALE) 100K benchmark for MaxMin TD Learning algorithm and the canonical temporal difference learning with $\epsilon$-greedy. Right:Median. Left: 80th Percentile.

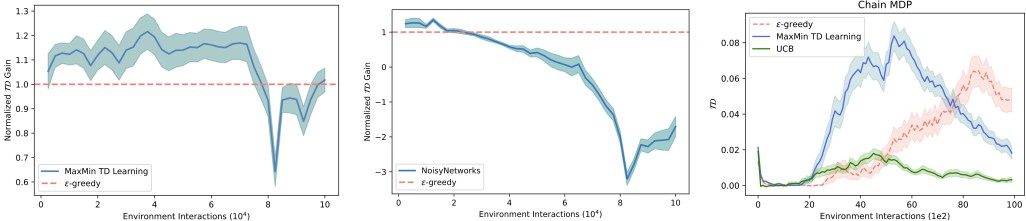

Figure 6: Left and Middle: Normalized temporal difference $\mathcal{TD}$ gain median across all games in the Arcade Learning Environment 100K benchmark for MaxMin TD Learning and NoisyNetworks. Right: Temporal difference $\mathcal{TD}$ when exploring chain MDP with Upper Confidence Bound (UCB) method, $\epsilon$-greedy and our proposed algorithm MaxMin TD Learning.

the temporal difference $\mathcal{TD}$

$$\mathbb{E}_{(s_t,a_t,s_{t+1})\sim\Lambda}\mathcal{TD}(s_t,a_t,s_{t+1}) = \mathbb{E}_{(s_t,a_t,s_{t+1})\sim\Lambda}[r(s_t,a_t) + \gamma \max_a Q_\theta(s_{t+1},a) - Q_\theta(s_t,a_t)].$$

The results reported in Figure 4 and Figure 6 further confirm the theoretical predictions made via Definition 3.2 and Proposition 3.4. In addition to the results for individual games reported in Figure 4, we compute a normalized measure of the gain in temporal difference achieved when using MaxMin TD Learning and plot the median across games. We define the normalized $\mathcal{TD}$ gain to be, Normalized $\mathcal{TD}$ Gain $= 1 + (\mathcal{TD}_{\text{method}} - \mathcal{TD}_{\epsilon\text{-greedy}})/(|\mathcal{TD}_{\epsilon\text{-greedy}}|)$, where $\mathcal{TD}_{\text{method}}$ and $\mathcal{TD}_{\epsilon\text{-greedy}}$ are the temporal difference for any given learning method and $\epsilon$-greedy respectively. The leftmost and middle plot of Figure 6 report the median across all games of the normalized $\mathcal{TD}$ gain results for MaxMin TD Learning and NoisyNetworks in the Arcade Learning Environment 100K benchmark. Note that, consistent with the predictions of Proposition 3.4, the median normalized temporal difference gain for MaxMin TD Learning is up to 25 percent larger than that of $\epsilon$-greedy. The results for NoisyNetworks demonstrate that alternate experience collection methods lack this positive bias relative to the uniform random action. The fact that, as demonstrated in Table 1, MaxMin TD Learning significantly outperforms noisy networks in the low-data regime is further evidence of the advantage the positive bias in temporal difference confers. The rightmost plot of Figure 6 reports $\mathcal{TD}$ for the motivating example of the chain MDP. As in the large-scale experiments, prior to convergence MaxMin TD Learning exhibits a notably larger temporal difference relative to the canonical baseline methods.

## 7 CONCLUSION

In our study we focus on the following questions in deep reinforcement learning: *(i) Is it possible to increase sample efficiency in deep reinforcement learning in a computationally efficient way with conceptually simple choices?, (ii) What is the theoretical motivation of our proposed perspective, minimizing the state-action value function in early training, that results in one of the most computationally efficient ways to explore in deep reinforcement learning?* and, *(iii) How would the theoretically motivated simple idea transfer to large scale experiments in high-dimensional state representation MDPs?* To be able to answer these questions we propose a novel, theoretically motivated method with zero additional computational cost based on following actions that minimize the state-action value function in deep reinforcement learning. We demonstrate theoretically that our method MaxMin TD

Learning based on minimization of the state-action value results in higher temporal difference, and thus creates novel transitions in exploration with more unique experience collection. Following the theoretical motivation we initially show in a toy example in the chain MDP setup that our proposed method MaxMin TD Learning results in achieving higher sample efficiency. Then, we expand this intuition and conduct large scale experiments in the Arcade Learning Environment, and demonstrate that our proposed method MaxMin TD Learning increases the performance on the Arcade Learning Environment 100K benchmark by $248\%$.

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
