# OpenReview forum: "Counterintuitive RL: The Hidden Value of Acting Bad"
_ICLR.cc/2025/Conference — Submitted to ICLR 2025_

### Official Review · Reviewer_qwcj · 2024-10-30

**Soundness:** 2
**Presentation:** 3
**Contribution:** 2
**Rating:** 3
**Confidence:** 3

**Summary:**

The authors propose MaxMin TD Learning, an algorithm that alternates between optimistic and pessimistic strategies for action sampling and Q-estimation updates. This approach addresses sample inefficiency in deep reinforcement learning (RL), though the method offers only incremental novelty. The authors provide both theoretical and empirical analysis, evaluating their method on the popular Atari benchmark.

**Strengths:**

The authors tested their approach using both DDQN and a more recent approach that represents the possible returns as distribution (QRDQN), which shows the flexibility of applying MaxMin TD Learning over a variety of different algorithms. Moreover, the tested environment is the well-known Atari benchmark, offering tasks with various characteristics. The paper is relatively easy to follow.

**Weaknesses:**

* *Marginal novelty*: The proposed method introduces limited novelty since exploring different selection criteria based on Q-estimations has been previously explored with ensembles [1, 2]. Additionally, similar work addressing the optimistic/pessimistic policy update trade-off exists using a more task-dependent strategy [3]. A Related Works section would help clarify where the proposed method advances existing literature. Furthermore, count-based exploration strategies should be referenced in the Background section for completeness.

* *Evaluation in high-dimensional MDPs*: The evaluation lacks depth, particularly concerning high-dimensional MDPs. MaxMin TD Learning is designed to enhance sample efficiency via exploration, yet it is compared against a standard $\epsilon$-greedy strategy, which performs well given a larger interaction budget and appropriately tuned decay factors. Limiting the interaction budget significantly impacts $\epsilon$ decay and policy performance, and it appears that the decay factor used here converges too rapidly to its minimum (see Q1). I would recommend including experiments with varying $\epsilon$ values, especially in the 200-million-frame setting. Additionally, while the 100k benchmark used the NoisyNetworks exploration strategy, it was absent in the 200-million-frame experiments.

* *Fair comparison*:  A more balanced comparison would be to benchmark MaxMin TD Learning against alternative approaches designed to enhance sample efficiency as seen in [4, 5]. The authors could emphasize the benefit of MaxMin TD Learning, such as enabling effective learning without requiring prior logged data or a guide policy, which could potentially lead to distribution shifts.

**General remarks**

- In the phrase “Thus, in high-dimensional complex MDPs…”, the citation of Kakade (2003) seems out of place, as deep reinforcement learning was developed later.

- The second question raised saying that the goal is to achieve a *zero cost* experience collection seems infeasible in the context of exploration since interactions with the environment have an inherently associated cost. I think the authors suggest *zero additional cost*

- I suggest having a single Reference section.

**References**

[1] Lan, Qingfeng, et al. "Maxmin q-learning: Controlling the estimation bias of q-learning." arXiv preprint arXiv:2002.06487 (2020).

[2] Agarwal, Rishabh, Dale Schuurmans, and Mohammad Norouzi. "An optimistic perspective on offline reinforcement learning." International conference on machine learning. PMLR, 2020.

[3] Moskovitz, Ted, et al. "Tactical optimism and pessimism for deep reinforcement learning." Advances in Neural Information Processing Systems 34 (2021): 12849-12863.

[4] Yao, Yao, et al. "Sample efficient reinforcement learning via model-ensemble exploration and exploitation." 2021 IEEE International Conference on Robotics and Automation (ICRA). IEEE, 2021.

[5] Uchendu, Ikechukwu, et al. "Jump-start reinforcement learning." International Conference on Machine Learning. PMLR, 2023.

**Questions:**

Q1: Could the authors clarify the decay rate for the hyperparameter $\textit{Exploration epsilon decay frame fraction}$? It seems that $\epsilon$ decreases every 0.8% of the total interaction budget. How was this value selected?

Q2: Could you provide more details on how NoisyNetworks was implemented in the experiments? Clarifying architecture choices and how this was selected would be useful, as it apparently allows for a range of configurations.

Q3: In Figure 1, constant 2 appears to balance exploration and exploitation in the UCB algorithm. Has this constant been optimized for the problem, and if so, could the results for varying values be shown? I'd like to see results for values such as [0.5, 1, 2, 3] as done for the epsilon value.

Q4: Could the authors clarify the intended interpretation of Figure 4 for the reader? What is exactly the meaning of TD error being stable or suffering from a drop during environment interactions? What are "high" negative or positive values in this case?

---

> ### Author Response · Authors · 2024-11-22
> **Author Response**
>
> Thank you for stating that our paper provides both theoretical and empirical analysis while our approach addresses sample efficiency, and further with the empirical analysis conducted in the well-known Atari benchmark, offering tasks with various characteristics.
>
> ---
>
> **1.** *Marginal novelty: The proposed method introduces limited novelty since exploring different selection criteria based on Q-estimations has been previously explored with ensembles [1, 2]. Additionally, similar work addressing the optimistic/pessimistic policy update trade-off exists using a more task-dependent strategy [3].*
>
> ---
>
> We believe you have substantial confusions regarding understanding the papers you refer to [1,2,3]. But this is not at all something that can be immediately addressed.
>
> The paper [1] employs multiple Q functions and minimizes over the Q functions. This is a completely different concept than our work in which our algorithm has one Q-network and minimization is done over actions for a given state not over multiple Q functions. This paper [2] is about offline reinforcement learning. This is a completely different concept/setup/subfield than our paper. Paper [3] learns a belief distribution over possible Q functions for actor-critic, again this is not relevant to our algorithm or our paper.
>
> [1] Qingfeng Lan, Yangchen Pan, Alona Fyshe, Martha White. Maxmin Q-learning: Controlling the Estimation Bias of Q-learning, ICLR 2020.
>
> [2] Rishabh Agarwal, Dale Schuurmans, Mohammad Norouzi. An Optimistic Perspective on Offline Reinforcement Learning, ICML 2020.
>
> [3] Ted Moskovitz, Jack Parker-Holder, Aldo Pacchiano, Michael Arbel, Michael I. Jordan. Tactical Optimism and Pessimism for Deep Reinforcement Learning, NeurIPS 2021.
>
> ---
>
> **2.** *Fair comparison: A more balanced comparison would be to benchmark MaxMin TD Learning against alternative approaches designed to enhance sample efficiency as seen in [4, 5]. The authors could emphasize the benefit of MaxMin TD Learning, such as enabling effective learning without requiring prior logged data or a guide policy, which could potentially lead to distribution shifts.*
>
> ---
>
> The paper [4] is about model based exploration, our paper is about increasing the temporal difference without any additional networks and models that learn additional metrics. The paper [5] is about using offline reinforcement learning and our paper is about off-policy reinforcement learning. These are completely different concepts and as such almost different subfields in reinforcement learning, thus there is no relevance of these studies to our paper.
>
> [4] Yao Yao, Li Xiao, Zhicheng An, Wanpeng Zhang, Dijun Luo. Sample Efficient Reinforcement Learning via Model-Ensemble Exploration and Exploitation, ICRA 2021.
>
> [5] Ikechukwu Uchendu, Ted Xiao, Yao Lu, Banghua Zhu, Mengyuan Yan, Joséphine Simon, Matthew Bennice, Chuyuan Fu, Cong Ma, Jiantao Jiao, Sergey Levine, Karol Hausman. Jump-Start Reinforcement Learning, ICML 2023.
>
> ---
>
> **3.** *Could the authors clarify the decay rate for the hyperparameter $\epsilon$? It seems that Exploration epsilon decay frame fraction decreases every 0.8% of the total interaction budget. How was this value selected?*
>
> ---
>
> Hyperparameters are set to the exact same values with prior studies to provide a fair and transparent comparison. These hyperparameters are also reported and explained in detail in the supplementary material.
>
> ---
>
> **4.** *Could you provide more details on how NoisyNetworks was implemented in the experiments? Clarifying architecture choices and how this was selected would be useful, as it apparently allows for a range of configurations.*
>
> ---
> The exact implementation of the prior studies [1] is used to provide consistent and fair comparison.
>
> [1] When to use parametric models in reinforcement learning?, NeurIPS 2019.
>
> ---
>
> **5.** *In Figure 1, constant 2 appears to balance exploration and exploitation in the UCB algorithm. Has this constant been optimized for the problem, and if so, could the results for varying values be shown? I'd like to see results for values such as [0.5, 1, 2, 3] as done for the epsilon value.*
>
> ---
>
> The value of the constant in UCB was actually in fact tuned to its best performing value which was 0.5 in these experiments. Other values in the interval $[0.5,3]$ performed worse, which results in slower convergence for larger values which can be found in the supplementary material.
>
> ---
>
> **6.** *Could the authors clarify the intended interpretation of Figure 4 for the reader? What is exactly the meaning of TD error being stable or suffering from a drop during environment interactions? What are "high" negative or positive values in this case?*
>
> ---
>
> In temporal difference learning targeting higher temporal difference, i.e. TD, will lead to faster learning. Hence, Figure 4 reports temporal difference results. The results reported in Figure 4 demonstrate that MaxMin TD learning leads to higher temporal difference throughout the training.

---

> > ### Comment · Reviewer_qwcj · 2024-11-25
> >
> > Thank you for your response.
> >
> > *We believe you have substantial confusions regarding understanding the papers you refer to [1,2,3].*
> >
> > I disagree with the assertion. Several components of the algorithm align conceptually with Offline RL methods, particularly in substituting the *argmax* operator with more conservative alternatives. Furthermore, I do not see any inherent limitation in applying these algorithms to online settings. For instance, paper [2] explicitly addresses this in Section 5.3 and provides supporting results in Figure 5. Similarly, [1] presents relevant results in Figure 3. However, I acknowledge that MaxMin TD Learning, as it encompasses both exploration and learning processes, belongs to a distinct category. This distinction, in my view, is the core issue with the paper. As noted by reviewer o2WG, the claim that this approach constitutes a **TD method** is assessed against an **exploration** strategy in a **low-data regime**, but the comparison is not fair under these circumstances.
> >
> > While I agree that [4, 5] are not suitable baselines for comparison due to their reliance on prior knowledge, I see no compelling reason to exclude comparisons with [1] and [2]. Specifically, it seems feasible to evaluate their learning processes both with and without the exploration strategy proposed by MaxMin TD Learning. Such a comparison could offer valuable insights into the performance of MaxMin TD Learning's exploration strategy. (1) If it demonstrates performance improvements over a naïve $\epsilon$-greedy approach, this would provide evidence of its effectiveness. (2) If demonstrate performance improvements employing the MaxMin TD Learning exploration strategy, this would provide evidence of the learning method's effectiveness.
> >
> > Lastly, I appreciate the inclusion of UCB results with varying constants in the appendix. However, for clarity, I recommend consolidating these results into a single plot (Appendix - 5.1, Figure 3), only showing UCB results. This would enhance the interpretability.

---

> > > ### Author Response · Authors · 2024-11-26
> > > **Author Response**
> > >
> > > Thank you for your response.
> > >
> > >
> > > Please observe that the paper [2] you reference compares QRDQN to their method (REM) as well as several other ensemble methods including bootstrapped DQN. Figure 5 of [2] plots the learning curves in the online setting for QRDQN, REM and bootstrapped DQN. The results reported in Figure 5 of [2] demonstrate that estimating the value-function distribution as QR-DQN does gives equivalent performance to estimating the value distribution via ensemble methods.
> > >
> > > The results reported in Figure 5 of [2] demonstrates that QRDQN performs identically to REM, and in our paper we indeed report results in the top performing algorithm, i.e. QRDQN, and these results reported in Figure 2 of our paper demonstrate that MaxMin TD learning substantially improves performance in the QRDQN case as well.
> > >
> > >  Furthermore, please see the supplementary material for the consolidated UCB results.

---

> > > > ### Comment · Reviewer_qwcj · 2024-11-26
> > > >
> > > > Thank you for your response. I will keep my score.

---

### Official Review · Reviewer_o2WG · 2024-11-01

**Soundness:** 1
**Presentation:** 2
**Contribution:** 2
**Rating:** 3
**Confidence:** 4

**Summary:**

This paper argues for an alternative exploration strategy to $\epsilon$-greedy for deep/approximate value-based RL methods (mostly those founded on Q-learning) in which instead of sampling uniformly at random with probability $\epsilon$, their approach samples actions based on $min_a Q(s, a)$. Algorithmically, the TD update rule of the basis algorithm remains intact. The authors argue that experiences generated by this method of acting have meaningful implications during experience replay/consolidation by TD methods (in particular, deep Q-learning family of algorithms). As such, the authors frame their proposal as a TD approach, in what they call MaxMin TD learning.

They examine the learning performance on a few tasks of the Atari suite (200M frames), full set of the Atari 100K benchmark, and an illustrative Chain MDP task. The key Atari results are based on the combination of their MaxMin TD learning with QRDQN (a distributional RL algorithm) in comparison with QRDQN with $\epsilon$-greedy, where MaxMin TD variant achieves higher AUC on both the Median and 80th Percentile aggregate measures.

**Strengths:**

**Interesting problem scenario:**
Considering exploration strategies that are directly useful for structural/temporal credit assignment is an interesting area to focus on in approximate/deep RL.

**Analysis tools around acting uniformly vs. Q-minimizing actions after parameter initialization:**
I found the approach of the propositions to analyze the impact of acting uniformly vs. taking the Q-minimizing action on the TD error to be interesting.

**Experimental testbeds:**
The choice of testbeds, ranging from a toy MDP problem to 100K and 200M Atari benchmarks is reasonable.

**Evaluation metrics:**
Reporting Median and 80% aggregate measures, using 5 seeds per each method in Atari for DQN-based methods, and reporting standard error of the mean return are all reasonable choices. However, statistical measures introduced by Agrawal et al. (2021) [arXiv:2108.13264] would have been a step up.

**Weaknesses:**

- **Framing the approach as a TD method, as opposed to an exploration strategy:**
Framing of the approach as a TD algorithm is not justifiable. A strategic exploration approach could facilitate credit assignment, but categorizing them as a TD approach is rarely ever useful in my view. The proposed method only touches experience generation and not experience consolidation and in this way, I see it as best described as a behavior/exploration strategy. Also, the baselines in question are Noisy Nets and $\epsilon$-greedy, which are both known as exploration/behavior strategies.

- **Propositions and proofs do not deliver** a full picture of what's going on, unlike the claims for theoretical foundations on par with those existing in tabular settings.

- The approach of only choosing actions from $max Q$ and $min Q$ can easily be shown to introduce bias in a simple counterexample. Say in a multiarmed bandit, action *a* is initialized to the minimal value at random (wrt. to the other initialized actions' values) but as it happens it's *true* Q value is lower than the initialized value. Let's assume also that action *b* is initialized to the maximal value at random (wrt. to the other initialized actions' values) and its corresponding *true* action value is higher than all other initialized actions' values. Note that, even if we use functional approximation (e.g. a neural network) to solve this problem, with parameters shared between the Q estimators for the various actions, it can easily end up being the case that no other actions are experienced during the course of training interactions. This would hold even despite the fact that neither of actions *a* and *b* would be the Q-minimizing or the Q-maximizing actions, respectively, wrt. the *true* Q function.

- Consider the example above again: The TD error reaches zero/near-zero for the Q-minimizing and Q-maximizing actions after a few updates (where Q is the current estimator and not the true Q-function). However, all other actions will have a much higher TD error since no updates is performed on them. This effectively shows that while the results of the Propositions in the paper could hold probabilistically assuming uniform initialization of the outputs of the Q-estimator network, they do not hold on a case by case setting nor do would they hold after several updates (after the uniformity of the outputs is no longer the case).

- Results on Atari 100K are significant, but not on Atari 200M experiments (especially given the fact that the plots for the latter are truncated earlier than 200M frames; e.g. StarGunner is truncated at 70M frames). This likely has ties to my argument above. MaxMin TD could help at the beginning of training (assuming settings like my counterexample occur less commonly in practice / in these tasks), but would not be able to reach higher final performances after enough training of a good baseline on the task.

- I think any benefit emerging from MaxMin TD could have ties to epistemic uncertainty minimization. I think discussions, detailed analysis, and comparisons with approaches directly purposed to do so (incl. bootstrapped DQN of Osband et al., 2016) would have been beneficial.

**Questions:**

1. Section 5 suggests that experiments were also done with Double DQN (with PER), however all I could find were learning curves for QRDQN, and Table 1 with results for DDQN. Generally, figuring out what results are based on which basis algorithm was challenging as I had to go back to the text several times without success. Could you please clarify what basis agent/algorithm each plot/table is based on. E.g. Figure 3's caption says "MaxMin TD Learning and canonical temporal difference learning in [ALE]...". The canonical TD learning method actually implies more of the TD($\lambda$) class of methods for prediction than a deep RL method such as DDQN or QRDQN. So please specify.

2. Why are the curves for Atari 200M truncated in some cases? (Could be beneficial to add the performance curves for the full length of the experiments.)

3. What was the reasoning behind choosing the specific subset of games for the Atari 200M experiments.

4. Can you comment on the counterexample that I've mentioned in the "Weaknesses" section? What is your view on it? (Perhaps experimenting with such a setting would be useful.)

Minor suggestions:
- Line 87: "MDP [...] contains continuous set of states"; I believe this intro is incorrect and also not applicable to the setting of this paper. In Atari and Chain MDP, states are in fact discrete. In Atari, pixel values are discrete, yielding a discrete combinatorial set of states.

- Line 89: The definition corresponds to the *expected* reward function.

- Line 90: The PMF-based definition of the policy does not hold fully for the continuous-state definition of the MDP. But this will be fine if Line 87 is changed to discrete set of states.

- Line 102: I believe the second expectation is mis-specified and in fact is not needed.

- Line 108: "In deep reinforcement learning, the state space or the action space is large enough that it is not possible to learn and
store the state-action values in a tabular form."; state-action spaces being large is not a property of DRL. I think better phrasing would be in this line: domains often tackled with DRL tend to have large state and/or action spaces.

-  Definition 3.3 seems to be formalized such that $\theta$ is a random variable of the expectation, but the wording seems to imply that $Q_\theta$ is a given.

- Would be good to have a visualization of the Chain MDP for ease of readability. Also, what was the number of states $N$?

- Number of environment interactions are not equal to the number of frames in Atari 2600 tasks, because of frame skipping of > 1 used. As such, the X axis labels should change to number of frames.

- The proposed approach is only compatible with discrete-action Q-based methods. That is to say, methods like DDPG cannot utilize it. I think it would be good to mention this somewhere.

---

> ### Author Response · Authors · 2024-11-22
> **Author Response Part I**
>
> Thank you for stating that our paper’s approach to analyze temporal difference and our proposal that is directly useful for structural/temporal credit assignment is interesting while our paper provides experimental analysis ranging from a toy MDP problem to 100K and 200M Atari benchmarks.
>
> ---
>
> **1.** *”Section 5 suggests that experiments were also done with Double DQN (with PER), however all I could find were learning curves for QRDQN, and Table 1 with results for DDQN. Generally, figuring out what results are based on which basis algorithm was challenging as I had to go back to the text several times without success. Could you please clarify what basis agent/algorithm each plot/table is based “*
>
> ---
>
> Figure 5 reports the results for Double DQN and Figure 2 reports results for QRDQN. This is also explained in Line 411 and 415. But we can further refer to it in multiple places to give a more smooth reading experience.
>
> ---
>
> **2.** *”Consider the example above again: The TD error reaches zero/near-zero for the Q-minimizing and Q-maximizing actions after a few updates (where Q is the current estimator and not the true Q-function). However, all other actions will have a much higher TD error since no updates is performed on them. This effectively shows that while the results of the Propositions in the paper could hold probabilistically assuming uniform initialization of the outputs of the Q-estimator network, they do not hold on a case by case setting nor do would they hold after several updates (after the uniformity of the outputs is no longer the case).”*
>
> ---
>
> Figure 4 demonstrates that the results of Propositions indeed hold not only after several updates but throughout the entire training up to convergence.
>
> ---
>
> **3.** *”Results on Atari 100K are significant, but not on Atari 200M experiments”*
>
> ---
>
> 200 million frame training in Tennis: Maxmin TD learning achieves the score of +5.448579367380699, the canonical methods obtain -6.71547619047, this is a 223.25% increase in performance.
>
> 200 million frame training in Gravitar: Maxmin TD learning achieves the score of 388.701902, and the canonical methods obtain 295.26349, this is a 31.64% increase in performance.
>
> 200 million frame training in Surround: Maxmin TD learning achieves the score of -6.6511238, and the canonical methods obtain -9.442219495, this is a 41.96% increase in performance.
>
> 200 million frame training in JamesBond: Maxmin TD learning achieves the score of 972.579366, and the canonical methods obtains 769.9060246, this is a 26.3% increase in performance.
>
> Similarly the increase in performance achieved by MaxMin TD learning in 200 million frame training persists across many games as reported in Figure 3, and stating that the increase in performance in 200 million frame training is insignificant is not a correct or fair assessment given the results provided in our paper.
>
> ---
>
> **4.** *”What was the reasoning behind choosing the specific subset of games for the Atari 200M experiments.”*
>
> ---
>
> Note that Figure 3 targets the games that are not part of the ALE 100K benchmark, as well as games which are part of the Arcade Learning Environment 100K benchmark to provide more comprehensive results. In particular, note that Gravitar, Surround, Bowling, StarGunner, and Tennis are not in the ALE 100K benchmark. Thus, these results provide more insight into what we can expect for the games that are not part of the ALE 100K benchmark. Furthermore, please note that some of these games in Figure 3 are also considered to be hard exploration games.
>
> ---
>
> **5.** *“Can you comment on the counterexample that I've mentioned in the "Weaknesses" section? What is your view on it? (Perhaps experimenting with such a setting would be useful.)”*
>
> ---
>
> As our extensive empirical analysis demonstrates, the contrived tabular counterexample you give does not seem to be an issue at all in deep reinforcement learning or in the tabular Chain MDP. This is because the learning dynamics itself, as our paper demonstrates, has sufficient randomness in it. Note that the standard convergence analysis of Q-learning requires only that every state-action pair is visited infinitely often. Hence, with sufficient noise in the learning dynamics this condition is also immediately satisfied via MaxMin TD, which indeed is the case as can be seen from the empirical analysis. If one is seriously concerned about contrived counterexample environments one can inject slight noise directly to MaxMin TD and immediately resolve the example.
>
>
>
> ---
>
> **6.** *”Why are the curves for Atari 200M truncated in some cases?”*
>
> ---
>
> In the cases where the policy converges earlier than 200 million frames, the figures were reported in zoomed versions to directly and clearly demonstrate the early convergence achieved.

---

> ### Author Response · Authors · 2024-11-22
> **Author Response Part II**
>
> ---
>
> **7.** *”Framing the approach as a TD method, as opposed to an exploration strategy: Framing of the approach as a TD algorithm is not justifiable. A strategic exploration approach could facilitate credit assignment, but categorizing them as a TD approach is rarely ever useful in my view. The proposed method only touches experience generation and not experience consolidation and in this way, I see it as best described as a behavior/exploration strategy. Also, the baselines in question are Noisy Nets and ϵ-greedy, which are both known as exploration/behavior strategies.”*
>
> ---
>
> The naming of our method is intended to capture the core intuition of our algorithm in which MaxMin TD learning increases the temporal difference for each transition.
>
> ---
>
> **8.** *”Propositions and proofs do not deliver a full picture of what's going on, unlike the claims for theoretical foundations on par with those existing in tabular settings.”*
>
> ---
>
> The propositions and proofs in our paper provide a theoretical analysis and justification for our proposed method MaxMin TD learning in which MaxMin TD learning selects transitions with higher temporal difference. Our empirical analysis in the the entire ALE 100K benchmark across different algorithms confirms this theoretical analysis and these mathematical predictions that indeed MaxMin TD learning does have higher temporal difference throughout the entire training, and that MaxMin TD learning indeed learns much faster. We do not anywhere in the paper mention that we provide theoretical foundations for the tabular setting, rather we mention that due to the concerns and the strong assumptions of the tabular settings, the tabular methods do not scale to deep reinforcement learning.
>
> ---
>
> **9.** *“I think any benefit emerging from MaxMin TD could have ties to epistemic uncertainty minimization. I think discussions, detailed analysis, and comparisons with approaches directly purposed to do so (incl. bootstrapped DQN of Osband et al., 2016) would have been beneficial.”*
>
> ---
>
> We do not see any clear connection to epistemic uncertainty minimization as in bootstrapped DQN.
> Bootstrapped DQN and related methods seek to maintain a distribution over value function estimates to explicitly learn and represent epistemic uncertainty about the true values.
> Our method does not employ additional deep neural networks to measure any sort of uncertainty, our method is based on increasing the temporal difference.
>
>
>
> ---
>
> **10.** *”Number of environment interactions are not equal to the number of frames in Atari 2600 tasks, because of frame skipping of > 1 used. As such, the X axis labels should change to number of frames.”*
>
> ---
>
> The x-axis reported is indeed correct. Indeed the number of environment interactions are not equal to the number of frames due to standard frame stacking. Please see the discussion in number of frames and number of environment interactions in Atari 2600 tasks in [1]. In particular please see Page 8 first paragraph, and furthermore please see Figure 3 in Page 9.
>
> [1] Hado van Hasselt, Matteo Hessel, John Aslanides. When to use parametric models in reinforcement learning?, NeurIPS 2019.

---

> ### Comment · Reviewer_o2WG · 2024-11-25
>
> Thanks for your response. Below I will comment on your responses.
>
> 1. Please do add that Fig. 5 is based on Double DQN in the caption.
>
> 2. Could you clarify how Fig. 4 shows this?
>
> 3.
> - Could you clarify how the 223.25% increase was computed? ("200 million frame training in Tennis: Maxmin TD learning achieves the score of +5.448579367380699, the canonical methods obtain -6.71547619047, this is a 223.25% increase in performance.")
>
> - Figure 3: Considering the confidence intervals, the results are not significant. The CIs are overlapping, with minor improvements in the mean performance levels. Also, let's consider for instance the game of Gravitar. Known results for DQN in this game reach a score of ~1300 (see, e.g., https://google.github.io/dopamine/baselines/atari/plots.html). However, the performance shown is ~300 for the baseline. When results are significantly below the known results, the variations in performance could come from slight implementation details or simply just insufficient number of trials. Could you clarify the number of seeds used and the basis agent used in Fig. 3? What basis implementation is being used?
>
> - Also, when we discuss performance improvements, it is very important to settle what we mean by performance first. It could be the mean performance at 200M frames, mean performance over the last 5M frames, the Area-Under-the-Curve (AUC), and so on. By truncating the X-axis at arbitrary points and using that performance for such assessments is simply scientifically incorrect.
>
> 4. Gravitar is the only hard exploration game in the set that I know of (see, e.g., Figure 4 of Oh et al. (2018) "Self-Imitation Learning"). However, I'm not familiar with Surround. But from what I can see in other papers, performance of -7.5 (close to what MaxMin TD Learning is obtaining) is on par with the performance of the random policy. Citing from Badia et al. (2020) "Agent57: Outperforming the human Atari benchmark": "For example, in the game Surround R2D2 achieves the optimal score while NGU performs similar to a random policy" where NGU's performance is provided in the mentioned paper's H.1 Table.
>
> 5. This unfortunately did not address my question. You mentioned: "If one is seriously concerned about contrived counterexample environments one can inject slight noise directly to MaxMin TD and immediately resolve the example." However, the question was: could you give me a solid reason not to be worried about having to inject additional noise?
>
> 6. When we discuss performance improvements, it is very important to settle what we mean by performance first. It could be the mean performance at 200M frames, mean performance over the last 5M frames, the Area-Under-the-Curve (AUC), and so on. By truncating the X-axis at different points and using that performance for such assessments is simply scientifically incorrect.
>
> 7. My grounding was the same as that you asserted. But could you clarify why an algorithm for action-selection should be categorized as a TD method?
>
> 8. No comment
>
> 9. No comment
>
> 10. What I'm saying is that the X-axis label should be 200M *frames*. But okay.

---

> > ### Author Response · Authors · 2024-11-25
> > **Author Response**
> >
> > ---
> >
> > **1.** *“Please do add that Fig. 5 is based on Double DQN in the caption.”*
> >
> > ---
> >
> > We will add the DDQN in the caption of Figure 5.
> >
> > ---
> >
> > **2.** *“Could you clarify how Fig. 4 shows this?”*
> >
> > ---
> >
> > In Figure 4, the solid lines report temporal difference for MaxMin TD, and the dashed lines report temporal difference for $\epsilon$-greedy, where the different colors correspond to different games. One can see in Figure 4 that the solid lines of each color are consistently above the dashed lines of the same color, indicating that TD is consistently higher for MaxMin TD than for the baseline methods throughout training. This demonstrates that the theoretical analysis provided in our paper, which shows that MaxMin TD learning will increase temporal difference, holds throughout training.
> >
> > ---
> >
> > **3.** *“Atari 200 million”*
> >
> > ---
> >
> > Below find the table reporting the area under the curve for 200 million frames training. Furthermore, note that MaxMin TD increases performance across all the games. Thus, the results demonstrate that MaxMin TD learning indeed increases the performance also in 200 million frame training.
> >
> >
> > |      Games          |    MaxMin    AUC        |  $\epsilon$-greedy AUC     |
> > |------------------------|-------------------------------|----------------------------------|
> > |  BankHeist         |   **164851.1469**         |    134149.1993971           |
> > |  StarGunner       |  **8161724.015**          |  7753506.86057              |
> > | Surround           |   **-1448.29515**           |    -1556.27495522           |
> > |  Gravitar            |    **28227.7661**            |     24133.36946                |
> > |  Tennis                |  **-1013.6794**             |       -1054.38959              |
> > |   Amidar             |    **204763.677**           |        195743.7222             |
> > |  JamesBond      |   **104197.2978**          |      89518.6469                |
> > |  Bowling             |   **8556.487**                  |      6980.375                  |
> >
> > ---
> >
> > **4.** *”Surround and Gravitar”*
> >
> > ---
> >
> > Surround is part of the Arcade Learning Environment benchmark, and the random policy score in Surround is -10.0 and the human score is 5.4. Thus the human normalized score in Surround for MaxMin TD learning is 21.7459% while human normalized score for baseline method is 3.62195%. Thus, MaxMin TD learning in fact achieves the $7\times$ the human normalized score of baseline methods.
> >
> > The baseline model is Double DQN and this is also explained in Line 372. The score achieved by Double DQN in Gravitar is 170.50 [1]. As reported by the original papers themselves the score achieved by DQN is 306.67. The results are not at all significantly below the known results. Please check the original papers that originally proposed these algorithms.
> >
> > [1] Hado van Hasselt and Arthur Guez and David Silver. Deep Reinforcement Learning with Double Q-learning, AAAI 2016.
> >
> > [2] Human-level control through deep reinforcement learning, Nature 2015.
> >
> > ---
> >
> > **5.** *”This unfortunately did not address my question. You mentioned: "If one is seriously concerned about contrived counterexample environments one can inject slight noise directly to MaxMin TD and immediately resolve the example." However, the question was: could you give me a solid reason not to be worried about having to inject additional noise?”*
> >
> > ---
> >
> > The results in high dimensional state observation MDPs across the entire benchmark with various algorithms indicate the contrived counter example is not an issue. The empirical analysis throughout the paper demonstrates that MaxMin TD learning not only converges, it obtains substantially higher scores.
> >
> > ---
> >
> > **6.** *”When we discuss performance improvements, it is very important to settle what we mean by performance first. It could be the mean performance at 200M frames, mean performance over the last 5M frames, the Area-Under-the-Curve (AUC), and so on. By truncating the X-axis at different points and using that performance for such assessments is simply scientifically incorrect.”*
> >
> > ---
> >
> > Please see response to item 2.
> >
> > ---
> >
> > **7.** *”My grounding was the same as that you asserted. But could you clarify why an algorithm for action-selection should be categorized as a TD method?”*
> >
> > ---
> >
> > The name of our algorithm is MaxMin TD learning because our algorithm maximizes the TD in every transition by minimizing the state-action value function, and furthermore we do not impose that our algorithm should be classified as a TD method. If the name of our algorithm causes a confusion for all we would be happy to rephrase the name of our algorithm.

---

> > > ### Comment · Reviewer_o2WG · 2024-11-25
> > >
> > > Thanks for your response. Some of my questions where not responded to (e.g. 3.1 and more), but for those responded I think I now have to take another look at the paper and other reviewers' comments to re-evaluate my assessment.
> > >
> > > The results in DQN and Double DQN are based on the deterministic Atari suite. I'm assuming your comparison is also on the deterministic version, and not the current best practices of using sticky actions to induce stochasticity in the transition dynamics?
> > >
> > > Regarding 7: Unfortunately this would require a rewrite of the paper, and not as simple as changing the algorithm name. But I will reassess the paper's exposition in its current status once again.
> > >
> > > Thanks for the AUC results. One important point was that, not only I needed clarification regarding the answers but also the paper needs revisions regarding its result reports. Number of seeds, name of basis algorithm easily attached to results (instead of "the canonical method"), non-truncated graphs, etc. If you can make these changes before the deadline for submitting rebuttal revisions, it would help me (and I'm sure other readers and reviewers) in re-assessing the work and its significance.

---

### Official Review · Reviewer_BH4h · 2024-11-04

**Soundness:** 2
**Presentation:** 3
**Contribution:** 3
**Rating:** 5
**Confidence:** 2

**Summary:**

The authors present a new algorithm which explores by choosing the worst action as estimated by the neural q function. They demonstrate the efficacy of this in low data regimes for double DWN and compare it to vanilla epsilon greedy based double DQN.

**Strengths:**

The paper provides a good set of experimental results in the low data regime. The method requires fairly simple changes to existing algorithms and it tends to improve performance while being so. The paper is largely well written and I was able to follow along easily.

**Weaknesses:**

I see a few important issues that need addressing before I can raise my score.

In **Proposition 3.4 and 3.6** you start with statements for state $s_t$, which is random variable corresponding to state at time $t$ but in the inequality on the RHS you somehow have $\mathcal D(s)$ for a fixed state $s$. I am unclear as to where this $s$ is coming from. Moreover, I believe this would significantly complicate the proofs because you will have to account for the time step or "loosen" the lower bound because you will have to take some kind of infimum.

While I am able to follow intuitively why you would benefit from taking the value minimizing action, I believe you should also include a **comment on the estimated regret** for this choice. It might be beneficial for a randomly initialized $Q$-function in a game setting but we must consider cases where large negative rewards are "harmful" to the agent. This is one of the reasons why minimizing regret is important to both theoreticians and practitioners.

In the experimental section I would be interested to see **comparison with two other papers**: "Exploration with random network distillation" and "Flipping Coins to Estimate Pseudocounts for Exploration in Reinforcement Learning". Former is based on quantifying how novel a data point is and the latter is directly related to optimistically choosing an action based on pseudo counts. You refer to the inability of doing count based exploration (as done in tabular setting) in your paper but these works are doing some form of counts based exploration. For reference, in lines 126-128 you write
>> incorporating these count-based methods in high-dimensional state representation MDPs requires substantial complexity including training additional deep neural networks to estimate counts or other uncertainty metrics

I would expect some comparison to how much better these more complex methods are.

**Questions:**

I see the following minor issues and make some auggestions:

Line 50: I wouldn't call $\epsilon$-greedy "naive and standard technique"

Line 96: comma at the end, not a full stop

Line 113: I believe you could use different subscript for $\theta$ to differentiate the gradient step from the environment time step.

Line 123: full stop at the end of eqn

Line 124: "a family of algorithms have been proposed based on counting state visitations" what are these algorithms? I would strongly recommend citing "R-max – A General Polynomial Time Algorithm for Near-Optimal Reinforcement Learning" and "Reinforcement Learning in Finite MDPs: PAC Analysis" here.

Line 155: Why is there an $s' \sim \mathcal T(s, \hat{a})$ in the first expectation? I don't see any dependence on $s'$ in the term inside the bracket. Same question for later versions of smoothness too.

Proof of Proposition 3.4: could you expand on the second inequality please?

Line 264: I am unclear on what is the "information gained" here? Is it in an information theoretic sense or in terms of optimizaing the loss?

---

> ### Author Response · Authors · 2024-11-22
> **Author Response**
>
> Thank you for stating that our paper is largely well written with a good set of experimental results while improving performance.
>
> ---
>
> **1.** *”In the experimental section I would be interested to see comparison with two other papers: "Exploration with random network distillation" and "Flipping Coins to Estimate Pseudocounts for Exploration in Reinforcement Learning". Former is based on quantifying how novel a data point is and the latter is directly related to optimistically choosing an action based on pseudo counts. You refer to the inability of doing count based exploration (as done in tabular setting) in your paper but these works are doing some form of counts based exploration. For reference, in lines 126-128 you write.”*
>
> ---
>
> Please note that both Random Network Distillation (RND) and Coin Flip Network utilize separate independent neural networks, besides the standard Q-Network to learn and predict independent metrics, i.e. RND utilizes additional recurrent neural networks. This is much different than the goal and objective of our paper which is to achieve sample efficiency across all games with zero additional cost. Our algorithm does not employ any extra networks to predict or separately learn anything.
>
> It is important to emphasize that neither of these prior papers on exploration in deep reinforcement learning claim to lead to faster learning. Rather both are only able to demonstrate an advantage over standard techniques in a few “hard exploration games,” and further require much larger numbers of environment interactions than what is focused on our paper, e.g. RND requires 1.97 billion frame training.
>
> In particular, RND (Random Network Distillation) is only tested in 6 hard-exploration games Montezuma’s Revenge, Pitfall, Solaris, Venture, and Private Eye. RND outperforms prior methods in only 3 of these games, and requires 1.97 billion frames of training to do so. Our results for faster learning via MaxMin TD instead require only 100K interactions (i.e. 400K frames) of training.
>
> Similarly, the flipping coins exploration method is tested in only 1 Atari game, Montezuma’s Revenge, where again the focus is on 200 million frame training. Thus, as these methods involve training additional neural networks to learn and estimate pseudocounts alongside standard RL methods, their advantages tend to only appear in certain hard-exploration settings where large numbers of frames of training are available in order to allow the uncertainty estimation networks to learn accurate estimates.
>
> Nonetheless, we still tested and added results in comparison to the CFN method from the paper “Flipping Coins to Estimate Pseudocounts for Exploration in Reinforcement Learning”  [1] in the supplementary material. The results demonstrate that MaxMin TD learning substantially outperforms both the canonical and recent methods.
>
> [1] Sam Lobel, Akhil Bagaria, George Konidaris. Flipping Coins to Estimate Pseudocounts for Exploration in Reinforcement Learning, ICML 2023.
>
> [2] Yuri Burda, Harrison Edwards, Amos Storkey, Oleg Klimov. Exploration by random network distillation, ICLR 2019.
>
> ---
>
> **2.** *In Proposition 3.4 and 3.6 you start with statements for state $s_t$, which is random variable corresponding to state at time $t$ but in the inequality on the RHS you somehow have $\mathcal{D}(s)$ for a fixed state $s$. I am unclear as to where this $s$ is coming from. Moreover, I believe this would significantly complicate the proofs because you will have to account for the time step or "loosen" the lower bound because you will have to take some kind of infimum.*
>
> ---
>
> Thank you for pointing this out. This is a typo and it should be $\mathcal{D}(s_t)$ not $\mathcal{D}(s)$.
>
> ---
>
> **3.** *“While I am able to follow intuitively why you would benefit from taking the value minimizing action, I believe you should also include a comment on the estimated regret for this choice. It might be beneficial for a randomly initialized $Q$-function in a game setting but we must consider cases where large negative rewards are "harmful" to the agent. This is one of the reasons why minimizing regret is important to both theoreticians and practitioners.”*
>
> ---
>
> In deep reinforcement learning rewards are typically normalized to the interval $[0,1]$ to stabilize training and allow these models to converge. Our method is designed for the deep reinforcement learning setting, rather than the online setting where every action may be associated with a cost, i.e. a large negative reward, and regret is the key metric. Thus, while minimizing regret online is a very important goal, both in theory and practice, it is not the objective of the significant body of work in deep reinforcement learning that our paper is a part of. We can definitely add a comment about regret.

---

> > ### Comment · Reviewer_BH4h · 2024-11-25
> > **Reply to the authors**
> >
> > Thank you for clarifying concerns 1 and 3 above. I see why in DRL you are not concerned with regret because it is in a game setting and you can normalize the scores. I also appreciate the comparison with CFN.
> >
> > I dont see the fix for $\mathcal D(s)$ in the latest version of paper though. Can you please fix that?
> >
> > Also, can you please answer why you have $s' \sim \mathcal T(s, \hat{a})$ on line 155-156 in the first expectation? I don't see where the transition is coming into the equation maybe I am missing something.

---

> > > ### Author Response · Authors · 2024-11-25
> > > **Reply to the Reviewer**
> > >
> > > Thank you very much for your response. On line 155-156 it is indeed not necessary to have $s’ \sim \mathcal{T}(s,a)$ for the first expectation. Now, we have fixed the typos you mentioned, and thank you again for pointing these typos out.

---

> > > > ### Author Response · Authors · 2024-12-02
> > > > **Thank You**
> > > >
> > > > We greatly appreciate you taking the time to review our paper. We trust that our response has addressed your questions. We wanted to ask if it would be possible for you to reassess your initial review in the light of our clarifications?
> > > >
> > > > Thank you again.
> > > >
> > > > Kind regards,
> > > >
> > > > Authors

---

### Official Review · Reviewer_EiPG · 2024-11-04

**Soundness:** 2
**Presentation:** 3
**Contribution:** 3
**Rating:** 6
**Confidence:** 4

**Summary:**

This work considers the problem of experience collection through behavior policy in RL. They argue the benefit of leveraging extremum actions to learn optimal policy and outlined an algorithm that collects and uses such samples. They theoretically show that how actions with minimum value could be helpful. Experimental validation of the approach has been conducted using the Atari game environment.

**Strengths:**

1. In my view, the main strength of this work lies in the presented theoretical assessment. It shows that the minimum-action value leads to higher temporal difference (TD) than random actions and the difference in the TD is equal to the disadvantage gap. Such finding reveals the underlying importance of the bad actions that may help in accelerate the learning which if often ignored.

2. This work nicely formalizes and defines the relevant concepts, and then gradually presents the core propositions. I have found the paper easy to follow. Also, the detailing of the propositions for both single and double Q-learning is helpful for the reader.

**Weaknesses:**

1. While the paper presents several experimental results on ALE, it lacks experiments across different benchmarks. It needs rigorous validation to uphold the claim. I would suggest adding more experiments on other benchmarks that are specially designed to assess exploration such as MiniGrid [1], Crafter [2].

2. Comparison with more recent and effective exploration techniques is missing. It would be interesting to see comparisons with Random Network Distillation [3] or curiosity-driven approaches [4] (which may require some adaptation).

3. Some part of the writing needs improvement. For example,
    - add more technical clarity such as in lines 291-292, please elaborate on what you intend to mean by "solely due to the experience collection".
    - minor grammatical issues such as "a fundamental theoretically well-motivated" -> "a fundamental and theoretically well-motivated" OR "a fundamental, theoretically well-motivated".
    - it is very hard to identify how Figures 2 and 5 differ. It would be helpful to the reader if you would add key information (the underlying architecture in this case) in the caption.

[1] Chevalier-Boisvert, Maxime, et al. "Minigrid & miniworld: Modular & customizable reinforcement learning environments for goal-oriented tasks." Advances in Neural Information Processing Systems 36 (2024).

[2] Hafner, Danijar. "Benchmarking the Spectrum of Agent Capabilities." International Conference on Learning Representations (2022).

[3] Yuri Burda, Harrison Edwards, Amos Storkey, Oleg Klimov. "Exploration by random network distillation", International Conference on Learning Representations (2019).

[4] Pathak, Deepak, et al. "Curiosity-driven exploration by self-supervised prediction." International Conference on Machine Learning (2017).

**Questions:**

1. It has been mentioned that "minimizing the state-action value function in early training ...". Does the algorithm considers actions with minimum value "only" in early training and does the value of $\epsilon$ in Algorithm 1 gradually reaches to zero? What is $e$ in algorithm 1?

2. Is there any motivating factor to cluster the games in figure 4 or is it just because of the value range?

---

> ### Author Response · Authors · 2024-11-22
> **Author Response**
>
> Thank you for stating that our paper reveals the underlying important components that accelerate learning, and nicely formalizes and defines the relevant concepts while providing a theoretical assessment, and then gradually presents the core propositions, and thank you for preparing a well-thought out review.
>
> ---
>
> **1.** *“It has been mentioned that "minimizing the state-action value function in early training ...". Does the algorithm considers actions with minimum value "only" in early training and does the value of $\epsilon$ in Algorithm 1 gradually reaches to zero? What is $\epsilon$  in algorithm 1?”*
>
> ---
>
> The values of $\epsilon$ are reported in the supplementary material. The values for $\epsilon$ are set to the exact same values with prior work to provide consistent and transparent comparison. Indeed, $\epsilon$ decay is a standard technique that gradually decreases the value of $\epsilon$.
>
> ---
>
> **2.** *“Is there any motivating factor to cluster the games in figure 4 or is it just because of the value range?”*
>
> ---
>
> Yes, indeed the clustering of the games in the graphs of Figure 4 is solely due to the value range and space efficiency.
>
> ---
>
> **3.** *"While the paper presents several experimental results on ALE, it lacks experiments across different benchmarks. It needs rigorous validation to uphold the claim."*
>
> ---
>
> We wanted to just briefly leave here that our paper provides empirical analysis in the low data regime across the entire Arcade Learning Environment benchmark and in the high data regime of Atari with Double DQN and QRDQN in the major canonical benchmark of deep reinforcement learning, and compares against the canonical methods $\epsilon$ greedy and NoisyNetworks while also providing results for count based methods, i.e. UCB in the Chain MDP.
>
> ---
>
> **4.** *"Some part of the writing needs improvement. For example, it is very hard to identify how figure 2 and 5 differ.”*
>
> ---
>
> Figure 2 reports results for QRDQN and Figure 5 reports results for DDQN. This is also explained in Line 411 and 415. But we can indeed further refer to them again to provide a better readability.

---

> > ### Comment · Reviewer_EiPG · 2024-11-26
> > **Reply to the authors**
> >
> > Dear Authors,
> >
> > Thanks for your response to my concerns and questions.
> >
> > While I am aware of the underlying mechanism of canonical $\epsilon$-greedy, I was more interested in knowing how exactly MaxMin TD is using $\epsilon$. My question was whether the algorithm starts solely looking at the min values ($\epsilon$ = 1.0) and then gradually transitions to solely optimizing the max value ($\epsilon$ = 0)  or keeps some min value optimization intact ($\epsilon$ = 0.01). Unfortunately, my question was not directly answered. Also, in the last part of the question, I intend to know more about $e$ which I believe is the rollout.
> >
> > I appreciate that you wanted to put some empirical analysis in the paper, however, as a proponent of an approach that aims to improve performance through better exploration, you should consider adding at least another benchmark that is designed to assess exploration such as MiniGrid [1], Crafter [2].
> >
> > I would say mentioning DDQN in the caption of Figure 5 will greatly facilitate the reader. Also, the discussion in lines 411-415, seems more tailored towards Figure 2.
> >
> > I would update my review including these suggestions, however, I would like to keep my current score.
> >
> > [1] Chevalier-Boisvert, Maxime, et al. "Minigrid & miniworld: Modular & customizable reinforcement learning environments for goal-oriented tasks." Advances in Neural Information Processing Systems 36 (2024).
> >
> > [2] Hafner, Danijar. "Benchmarking the Spectrum of Agent Capabilities." International Conference on Learning Representations (2022).

---

### Meta-Review · Area_Chair_A5RP · 2024-12-21

**Metareview:**

This paper introduces a counterintuitive notion that instead of exploring uniformly randomly, exploring with minimum-value action helps enlarge the temporal difference error and hence benefits learning. Justification in terms of theory is given. The approach can be well formulated in terms of max-min optimization. One issue I find is that following the minimum-value action might be quite harmful in some cases.

The learning part of the pseudocode in Algorithm 1 is confusing. I believe the updates are made simply based on the drawn samples. The statement “TD receives update with probability ϵ:” serves as an explanation of what’s happening instead of an if-else mechanism that needs to be implemented, an impression of which is created by this block of statement and it may confuse readers. This explanation should be moved out of the pseudocode.

The implication of their results in the asymptotic sense and comparison to a range of baseline exploration strategies and benchmark tasks would make the algorithm much stronger and acceptable in the next submission.

**Additional Comments On Reviewer Discussion:**

The reviewers were mixed about the work. While they appreciated the counterintuitive nature of the result, the work overall seems rather incomplete, some of which I already pointed out above. For example, a reviewer brought up the issue of potential harm in taking minimum-value actions. I found the response by the authors rather confusing: “Our method is designed for the deep reinforcement learning setting, rather than the online setting where every action may be associated with a cost, i.e. a large negative reward, and regret is the key metric. Thus, while minimizing regret online is a very important goal, both in theory and practice, it is not the objective of the significant body of work in deep reinforcement learning that our paper is a part of.” Deep RL can be and is used online. There is no agreed-upon notion that deep RL is mainly concerned with being applicable in manipulable games. This concern should be addressed by acknowledging that it can potentially be highly problematic.

---

### Decision · Program_Chairs · 2025-01-22

Reject